# A Deformable Shape Model for Automatic and Real-Time Dendrometry

Lucas A. Wells *,† and Woodam Chung

Department of Forest Engineering, Resources and Management, College of Forestry, Oregon State University, Corvallis, OR 97331, USA; woodam.chung@oregonstate.edu
* Correspondence: lucas@silvxlabs.com
† Current address: Silvx Labs, Missoula, MT 59802, USA.

**Abstract:** We present a stereo image-based algorithm for tree stem diameter measurement and form analysis. The algorithm uses planar parametric curves to represent two-dimensional projections of tree stems in stereo images. The curves evolve according to an energy formulation based on the gradients of the images and inductive priors related to biomechanics and morphology of tree stems. After energy minimization, the curves are reconstructed to three dimensions, allowing for diameter measurements at any point along the height of the stem. We describe the algorithm and report the validation test results comparing predicted diameter measurements to external observations. Our findings demonstrate that the algorithm can automatically estimate diameters for trees within 20 m of the camera with an error of 5.52%. We highlight how this method can aid product value optimization through taper analysis and sweep or crook detection. A run-time analysis shows that the algorithm can estimate dendrometric variables for ten trees simultaneously at 15 frames per second on a consumer-grade computer. Furthermore, we discuss the opportunity to produce training data for machine learning algorithms that generalize across domains and eliminate the need to manually tune parameters.

**Keywords:** diameter measurement; stem form analysis; stereo camera; computer vision; active contour model





## 1. Introduction

In forestry, decision-making often hinges on measurements obtained from individual trees. Given that such decisions can substantially influence both environmental and economic outcomes at broad scales, minimizing measurement errors and biases, and securing ample data to sufficiently decrease estimation errors are essential. Yet, due to the vast spatial reach of forests, the cost-effective collection of measurements remains a formidable challenge, making the reconciliation of these conflicting demands a central issue in forest inventory.

The emergence of contemporary sensor technologies, including digital photography and laser ranging, has considerably augmented our capacity to rapidly amass large data sets, often at decreased costs. These technologies have been employed in aerial surveys aimed at quantifying forest structure [1], estimating attributes [2–4], and mapping stems [5,6]. While aerial surveys provide extensive spatial information, they often depend on allometric correlations between sub-canopy attributes, such as diameter, and observations like tree height and crown width. However, allometric models utilizing aerial observations may only yield accurate predictions for select sub-canopy dendrometric variables of interest, falling short in providing data related to under-story density and composition. Consequently, there is a burgeoning interest in harnessing sensor technologies in terrestrial surveys to augment data collection efficiency and consistency.

One of the earliest instances of terrestrial remote sensing applications in dendrometry was introduced by Shelbourne and Namkoong [7]. They utilized two photographs of a tree

stem, captured at 90-degree angles, to compute stem straightness in three dimensions (3D) by using the center-line of the two-dimensional (2D) projections. This approach was later refined by Hapca et al. [8], who integrated digital photography and extended the method to include stem taper profiles. They showcased the potential to automatically classify 3D stem reconstructions into form classes using their method [9].

Presently, terrestrial remote sensing research is primarily focused on the application of light detection and ranging (LiDAR) systems. Researchers have established the potential for constructing highly precise and intricate tree models [10,11] and automating stem detection and measurement [12–14] using LiDAR. However, despite the efficacy and accuracy of LiDAR systems in various forestry applications, their high cost and data processing demands render them impractical for certain uses.

The recent development of sophisticated computer vision algorithms and the availability of publicly accessible software has incited renewed research interest in photogrammetric approaches to dendrometry, specifically in the application of multi-view stereo and structure-from-motion (MVS-SFM) algorithms. This method involves converting a sequence of 2D images into a 3D point cloud, akin to data captured by a LiDAR sensor. The initial demonstrations of MVS-SFM application to dendrometry by [15,16] utilized a camera mounted on an unmanned aerial system (UAV) to acquire images of tree stems from various perspectives, reconstructing 3D point clouds and surface models. Subsequent studies by [17,18], using handheld cameras, discovered that diameter at breast height (DBH) estimations from MVS-SFM point clouds matched those estimated using terrestrial LiDAR. Later investigations into MVS-SFM-derived point clouds centered on detailed accuracy assessments [19], amalgamating aerial and terrestrial data [20], multi-camera configurations [21], and taper modeling [22].

To date, all applications of MVS-SFM for dendrometry follow a general workflow that includes data acquisition, offline processing for point cloud generation, and analysis of the point cloud through manual or automated measurements. While this approach offers satisfactory accuracy for most forestry applications and requires relatively inexpensive equipment, the computational expense of offline processing for point cloud generation can be significant, often taking hours or even days to complete. As such, in scenarios where immediate stem measurements are required, MVS-SFM may prove impractical. One such instance is markless prescriptions, wherein a description of the silvicultural treatment is provided to the operator of a harvesting machine tasked with selecting the trees to be cut. Such prescriptions frequently specify diameter limits, species preferences, and spacing requirements. In this context, a terrestrial remote sensing system that can deliver real-time, reliable measurements of trees within close proximity could prove advantageous for operators.

This paper presents a novel method for photogrammetric dendrometry, differing significantly from recent multi-view stereo and structure-from-motion (MVS-SFM) approaches. Our algorithm, based on fundamental photogrammetric principles, provides 3D reconstructions of tree stems and enables quantification of diameter, taper, sweep, and crook. This approach echoes earlier methods for analyzing tree stems in photographs [7] and incorporates the core principles of the active contour model presented in [23]. The inputs to the algorithm are the images from a calibrated stereo camera and the output of a bounding box object detector, trained on tree stems [24]. The algorithm automatically localizes the occluding contours of the tree stems within the bounding box in both the left and right stereo images, reconstructs the stems, and maps their positions to the object coordinate frame. Running in real-time on a modest graphics processing unit (GPU), the algorithm can provide reliable measurements for trees up to 20 m from the camera.

Aside from its direct applications in dendrometry, the approach presented here can also facilitate the creation of a training dataset for machine learning (ML) algorithms capable of segmenting tree stems in digital images. Although our method effectively localizes occluding contours in stereo images, the parameters of the algorithm require manual tuning specific to the problem domain. An ML-based approach, such as a region-based

convolutional neural network (RCNN) [25], could offer a more generalizable and scalable solution. By using the 2D segmentation masks generated by our algorithm as training data for an ML-based segmentation model, we can develop a more robust system capable of segmenting tree stems under various environmental conditions and among different tree species. Consequently, this paper does not only present a novel photogrammetric approach to dendrometry but also lays the groundwork for future research that can integrate our work with state-of-the-art ML algorithms.

In this paper, we provide a comprehensive description of the algorithm, detailing the model, energy formulation, and optimization technique. We also delve into various algorithmic properties, showcase the results of a diameter measurement validation study, and discuss potential approaches to assess stem form and optimize product value from wood property inference. Figure 1 visually illustrates the major components of our proposed method.

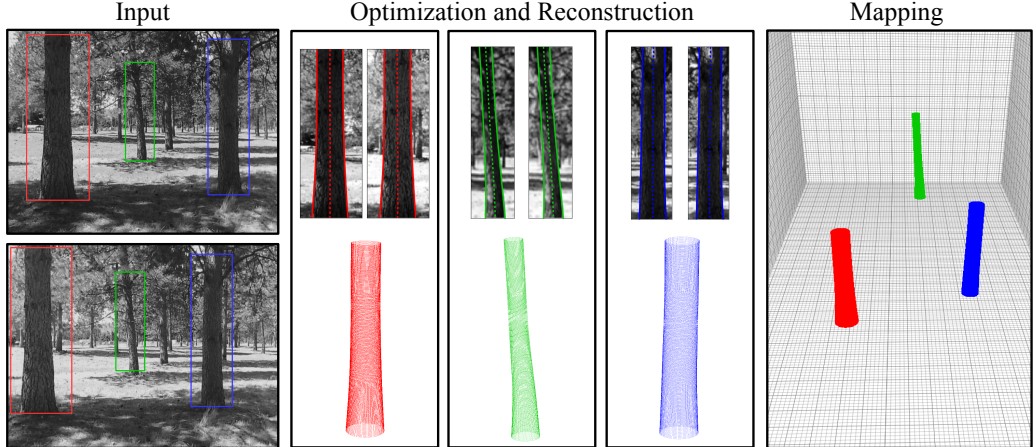

**Figure 1.** Outline of our proposed method. Given a stereo pair and tree stem detection boxes, the algorithm localizes the edges of the tree in the left and right stereo images and reconstructs their 3D shape. The stems are then mapped to the object coordinate frame [26].

## 2. Algorithm Description

We are given a stereo-rectified and row-aligned image pair, $\{\mathcal{I}^L(x,y), \mathcal{I}^R(x,y)\}$, acquired from a stereo camera in the fronto-parallel configuration. Images are represented in function notation, $\mathcal{I} : \Omega \to \mathbb{R}$, where $\Omega \subset \mathbb{R}^2$ is the set of pixel coordinates in the image plane. We are also given the output of an object detector trained to detect tree stems [24]. The object detector outputs $n$ bounding boxes, $\{\Phi_1, \Phi_2, \ldots, \Phi_n\}$, where each $\Phi_i \subset \Omega$. For example, we use the notation $\mathcal{I}_{\Phi_i} : \Phi_i \to \mathbb{R}$ to specify an image function within the domain of some arbitrary bounding box and $\mathcal{I} : \Omega \to \mathbb{R}$ to specify the entire image. We use upright, bold Greek letters to represent vector-valued functions in $\mathbb{R}^2$, e.g., $\boldsymbol{\chi} : \mathbb{R} \to \mathbb{R}^2$, bold capital letters for matrices, e.g., $\mathbf{A}$, and lowercase bold letters for vectors, e.g., $\mathbf{x}$. We use the notation $\|\cdot\|$ as shorthand for $\|\cdot\|_2$, the Euclidean norm. Other notational conventions are apparent in context.

### 2.1. Deformable Stem Model

We use a set of parametric curves to represent the occluding contours of a tree stem projected on the image plane. We assume that there exists a curve of symmetry such that for all points on the curve, the unit normal, multiplied by some scalar, i.e., the radius, can be rotated about the tangent axis to the edge of the tree. Such a curve does not exist in 3D space, as the cross-section of a tree stem can only be approximated by a circle. However, this assumption holds for the two-dimensional case, and we exploit this symmetry to construct a shape model for tree stems.

Consider two curves, $\{\boldsymbol{\lambda} : \mathbb{R} \to \mathbb{R}^2, \boldsymbol{\rho} : \mathbb{R} \to \mathbb{R}^2\}$, representing the left and right contours of a tree stem projected in the $x, y$ plane. We can construct these contours using

the curve of symmetry, $\chi : \mathbb{R} \to \mathbb{R}^2$, and a taper function, $\tau : \mathbb{R} \to \mathbb{R}^+$, representing the change in radius along the stem. The left and right contours are defined as

$$\boldsymbol{\lambda}(s) = \boldsymbol{\chi}(s) - \tau(s)\hat{\mathbf{n}}(s) , \tag{1a}$$

$$\boldsymbol{\rho}(s) = \boldsymbol{\chi}(s) + \tau(s)\hat{\mathbf{n}}(s) . \tag{1b}$$

where $s$ is the parameterization, $s \in [0,1]$, and the unit tangent and unit normal vectors of $\boldsymbol{\chi}(s)$ are given by

$$\hat{\mathbf{t}}(s) = \frac{\boldsymbol{\chi}'(s)}{\|\boldsymbol{\chi}'(s)\|} , \tag{2a}$$

$$\hat{\mathbf{n}}(s) = \frac{\hat{\mathbf{t}}'(s)}{\|\hat{\mathbf{t}}'(s)\|} , \tag{2b}$$

Modeling the curves as presented requires an arbitrary number of parameters that depend on the level of discretization. In practice, discretization is often performed at the pixel level, resulting in hundreds of parameters that significantly increase computational demands during iterative optimization presented in Section 2.5. To address this challenge, we introduce a layer of abstraction to model tree stems that allows for efficient and compact representations without sacrificing the generality of the approach, even for instances of highly varied forms.

Let $\mathcal{X} = \{(x_i, y_i)\}_{i=1}^n$ denote an ordered set of coordinates in the plane. We use vectors to describe first and second entries independently, i.e., $\mathbf{x} = (x_1, x_2, \ldots, x_n)^\mathsf{T}$ and $\mathbf{y} = (y_1, y_2, \ldots, y_n)^\mathsf{T}$. The coordinates in $\mathcal{X}$ correspond to $n$ points on the curve of symmetry describing a tree stem. We fix the elements of $\mathbf{y}$ to be equally spaced along the height of the stem by

$$y_i = \frac{i-1}{n-1}h \qquad \forall\, i = \{1, 2, \ldots, n\} , \tag{3}$$

where $h$ is the height of the stem, equivalent to the number of pixel rows in the bounding box. We treat the vector $\mathbf{x}$ as a free parameter that can take on arbitrary values to represent an instance of the model. A curve can be constructed by interpolating $\mathcal{X}$ with a cubic spline interpolant $\mathcal{S}(t) \in \mathcal{C}^2[y_1, y_n]$ of the form

$$\mathcal{S}(t) = \begin{cases} \mathcal{S}_1(t), & y_1 \le t \le y_2 \\ \mathcal{S}_2(t), & y_2 \le t \le y_3 \\ \;\;\vdots \\ \mathcal{S}_{n-1}(t), & y_{n-1} \le t \le y_n \end{cases} , \tag{4}$$

where each $\mathcal{S}_i(t)$ is a polynomial of degree three. We require that the interpolant passes through $\mathbf{x}$, i.e., $\mathcal{S}(y_i) = x_i$ for all $\{i\}_1^n$, and constrain $\mathcal{S}''(y_1) = \mathcal{S}''(y_n) = 0$ to specify natural boundary conditions. The coefficients of the polynomials can be solved in $\mathcal{O}(n)$ using the tridiagonal matrix algorithm [27]. Using $\mathcal{S}_\mathbf{x}(t)$ to denote the spline function that depends on the parameter vector $\mathbf{x}$, the curve of symmetry can now be defined as

$$\boldsymbol{\chi}(s) = (\mathcal{S}_\mathbf{x}(hs), hs)^\mathsf{T} , \tag{5}$$

where $s$ is the parameterization, $s \in [0,1]$.

Now that the curve of symmetry is specified, we define the taper curve in a similar manner. Let $\mathcal{R} = \{(r_i, y_i)\}_{i=1}^n$ be a set of $n$ coordinates where the first entry elements, $\mathbf{r} = (r_1, r_2, \ldots, r_n)^\mathsf{T}$, specifies the radius of the stem at each corresponding second entry $\mathbf{y}$, and each $y_i$ is defined by Equation (3). Using $\mathcal{R}$ we construct a cubic spline, $\mathcal{S}_\mathbf{r}(t)$, satisfying the same conditions as for $\mathcal{S}_\mathbf{x}(t)$, but in terms of the free parameter $\mathbf{r}$. Following from

Equation (5), a taper function describing the change in radius along the length of the stem is given by

$$\tau(s) = \mathcal{S}_{\mathbf{r}}(hs), \quad s \in [0,1].\tag{6}$$

Computing the unit tangent and unit normal vectors as shown in Equation (2) with respect to $\chi(s)$, we can use Equation (1) to define the curves representing the left and right contours of the stem, $\lambda(s)$ and $\rho(s)$; however, now they depend only on the parameter set $\{\mathbf{x}, \mathbf{r}\}$.

Figure 2 illustrates five instances of the Deformable Stem Model. Each instance is constructed using a different number of control points, $n$. As the number of control points increases, the number of parameters required to specify the model increases linearly by $2n$. Although an instance with a larger $n$ can represent more complex stem forms, all stems we encountered during experimentation were sufficiently represented with 5–7 control points.

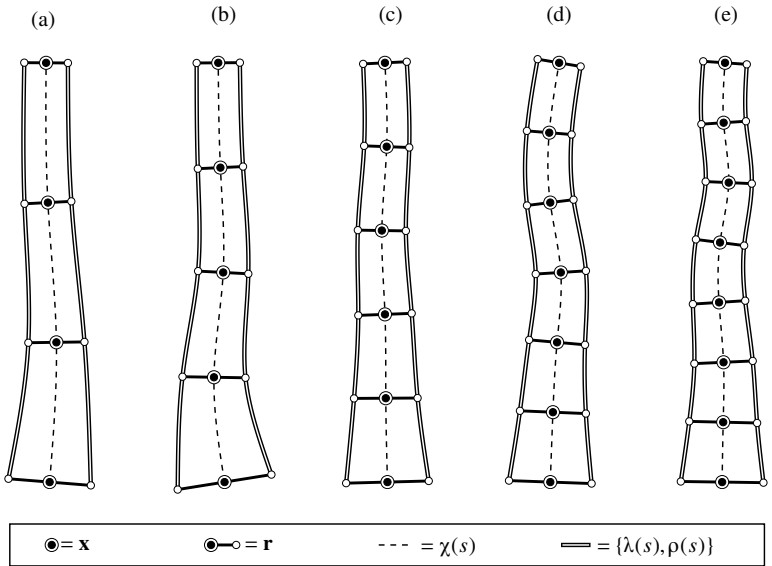

**Figure 2.** Deformable Stem Model: Five instances of the Deformable Stem Model each with a different number of control points ($n$). (**a**) $n = 4$; (**b**) $n = 5$; (**c**) $n = 6$; (**d**) $n = 7$; (**e**) $n = 8$.

Extension to 3D

To extend our model to the object coordinate frame, $\mathbb{R}^3$, we can construct two sets of curves in stereo such that one corresponds to the left camera and the second to the right. We use superscripts $L$ and $R$ on the functions to denote a curve in the left and right image, respectively. Assuming that the stereo pair is rectified and row-aligned, the curves can be back-projected to the object coordinate frame via triangulation. First, we compute the inverse disparity, $\delta : \mathbb{R} \to \mathbb{R}^+$, given by

$$\delta(s) = \left| \chi_x^L(s) - \chi_x^R(s) \right|^{-1},\tag{7}$$

where the subscript on the functions index the corresponding position in the output vector. Using the inverse disparity and the stereo calibration parameters, the curve of symmetry in the left camera coordinate frame is representing in object coordinates by the function $X : \mathbb{R} \to \mathbb{R}^3$, defined as

$$X(s) = \begin{pmatrix} \left( \chi_x^L(s) - c_x \right) \delta(s) b \\ \left( \chi_y^L(s) - c_y \right) \delta(s) b \\ f \delta(s) b \end{pmatrix},\tag{8}$$

where $(c_x, c_y)$ is the principal point of the left image, $f$ is the focal length in pixels and $b$ is the baseline distance between the optical axes of the left and right cameras. The taper function is scaled to object coordinates by

$$T(s) = \tau^L(s)\delta(s)b,\tag{9}$$

where $T : \mathbb{R} \to \mathbb{R}^+$. Given the back-projected curve of symmetry, $X(s)$, and the scaled taper function, $T(s)$, we can define a parametric surface function, $\Lambda : \mathbb{R}^2 \to \mathbb{R}^3$, based on Rodrigues' rotation formula to reconstruct the tree stem to 3D:

$$\Lambda(s, \theta) = X(s) + T(s)\left(\mathbf{I} + \sin\theta \left[\hat{\mathbf{t}}(s)\right]_\times + \left(1 - \cos\theta\right)\left[\hat{\mathbf{t}}(s)\right]_\times^2\right)\hat{\mathbf{n}}(s),\tag{10}$$

where $s$ and $\theta$ are the parameterizations, $s \in [0, 1]$ and $\theta \in [0, 2\pi]$, $\mathbf{I}$ is a $3 \times 3$ identity matrix, $\hat{\mathbf{t}} : \mathbb{R} \to \mathbb{R}^3$ and $\hat{\mathbf{n}} : \mathbb{R} \to \mathbb{R}^3$ are the unit tangent and normal vector functions of $X(s)$, and $[\hat{\mathbf{t}}(s)]_\times$ is a skew symmetric matrix of the unit tangent vector defined as

$$\left[\hat{\mathbf{t}}(s)\right]_\times = \begin{pmatrix} 0 & -\hat{t}_z(s) & \hat{t}_y(s) \\ \hat{t}_z(s) & 0 & -\hat{t}_x(s) \\ -\hat{t}_y(s) & \hat{t}_x(s) & 0 \end{pmatrix}.\tag{11}$$

### 2.2. External Energy

Active contour models use an external energy to attract contours to desirable features in the image. External energy is derived from the input image in such a way that minimum values correspond to the desirable features. Our objective is to localize the edges of tree stems, so we will formulate the external energy to be minimal near vertical edges in the input image. We also introduce a secondary external energy to add robustness to local minima during curve evolution.

#### 2.2.1. Edge Map and Gradient Flow

We define an edge map $\mathcal{I}_x : \Phi_i \to \mathbb{R}^+$ by differentiating the image $\mathcal{I} : \Phi_i \to \mathbb{R}$ w.r.t $x$ and taking the absolute value,

$$\mathcal{I}_x(x, y) = \left|\frac{\partial \mathcal{I}}{\partial x}\right|.\tag{12}$$

In practice, it is recommended to denoise the image prior to computing gradients via Gaussian smoothing or bilateral filtering. We avoid computing the gradient vector $\nabla \mathcal{I}$ (differentiating $\mathcal{I}$ w.r.t $x$ and $y$) as edges that represent the boundaries of the stem will only have strong gradients in the $x$ direction while gradients in the $y$ direction could introduce spurious forces that adversely influence the convergence of the contours.

Differentiating the edge map, $\mathcal{I}_x(x, y)$, w.r.t $x$ gives a function where the zero-crossings correspond to edges in the image. If we consider this as a functional taking a parametric curve as an input, then the sign and magnitude of the scalar values returned by the functional indicate the direction and speed in which to move the curve. This, however, assumes that the curve, at some point during evolution, is within the capture range of a boundary in the edge map, which is generally an unrealistic assumption. Xu and Prince [28] present an alternative formulation for external energy, called gradient vector flow (GVF), that overcomes issues with insufficient capture range by filling in regions where gradients are not present, effectively increasing the capture range of the edge map.

GVF is a vector field $\mathbf{v}(x, y) = (u(x, y), v(x, y))^\mathsf{T}$ that minimizes the energy

$$\mathcal{E}_{uv} = \iint \eta(u_x^2 + u_y^2 + v_x^2 + v_y^2) + \|\nabla \mathcal{I}_x\|^2\|\mathbf{v} - \nabla \mathcal{I}_x\|^2 \, \mathrm{d}x \, \mathrm{d}y,\tag{13}$$

where $\eta$ is a regularization parameter that controls the relative importance of the first and second terms in the integrand. Since the Deformable Stem Model considers updates only in the $x$ direction, we simplify the expression by omitting the $y$-component of the vector field yielding

$$\mathcal{E}_u = \int \eta(u_x^2 + u_y^2) + \|\mathcal{I}_{xx}\|^2 \|u - \mathcal{I}_{xx}\|^2 \, dx \,, \tag{14}$$

where $\mathcal{I}_{xx}$ is defined as

$$\mathcal{I}_{xx}(x, y) = \frac{\partial \mathcal{I}_x}{\partial x} \,. \tag{15}$$

As shown by Xu and Prince [28], $u$ can be found by expressing Equation (14) as a Euler equation,

$$\eta \nabla^2 u - (u - \mathcal{I}_{xx}) \mathcal{I}_{xx}^2 = 0 \,, \tag{16}$$

where $\nabla^2$ is the Laplacian operator, and then solving via iteration by treating $u$ as a function of time,

$$u_{t+1}(x, y) = \eta \nabla^2 u_t(x, y) - (u_t(x, y) - \mathcal{I}_{xx}(x, y)) \mathcal{I}_{xx}(x, y)^2 \,. \tag{17}$$

Through experimentation, we found $\eta = 0.2$ to be a sufficient setting for the regularization parameter. We use the function $\mathcal{G} : \mathbb{R}^2 \to \mathbb{R}$ to represent the steady-state solution to $u_t(x, y)$. We refer to $\mathcal{G}(x, y)$ as the gradient flow force function to indicate that it is not a vector field as originally proposed by Xu and Prince [28]. $\mathcal{G}(x, y)$ returns a signed scalar value where the sign denotes the direction of the force and the value denotes the magnitude of the force queried at the position variables $x$ and $y$.

### 2.2.2. Contraction Energy

The edge map $\mathcal{I}_x$ has the property that large values correspond to abrupt intensity changes in the horizontal direction. Ideally, large values would represent the boundaries of the stem while the rest of the domain exhibits small values. However, $\mathcal{I}_x$ is likely to contain large values elsewhere; large values might exist in the background of the image caused by adjacent stems, branches, or other features in the image. We formulate a contraction energy based on the stereo disparity map to make the contours robust to local minima that do not correspond to gradients along the tree stem boundary. We compute an integer stereo disparity map for the entire image domain, $\mathcal{D} : \Omega \to \mathbb{N}$, using semi-global matching [29]. We use a real-time GPU implementation of the semi-global matching algorithm presented by [30].

Let $\mathcal{B} : \mathbb{R}^2 \to \{0, 1\}$ be a logical function defined as

$$\mathcal{B}(x, y) = \begin{cases} 1 & \text{if } d - \theta \leq \mathcal{D}_\Phi(x, y) \leq d + \theta \\ 0 & \text{otherwise} \end{cases} \,, \tag{18}$$

where $\mathcal{D}_\Phi : \Phi_i \to \mathbb{N}$ is the stereo disparity map within the domain of the bounding box, $\Phi_i$. The threshold value $d$ can be readily found by computing the histogram of $\mathcal{D}_\Phi(x, y)$ and taking $d$ as the argument maximum of the histogram. The range value $\theta$ is the expected range above and below $d$ used to capture pixels of the tree stem projected on disparity planes other than $d$, typically 3 to 5. Next, we compute a row-wise distance transform, $\mathcal{W} : \mathbb{R}^2 \to \mathbb{R}$, given by

$$\mathcal{W}(x, y) = \max_{u : \mathcal{B}(u, y) = 1} -\frac{1}{2}(u - x)^2 \,. \tag{19}$$

The form of Equation (19) becomes apparent when taking its derivative w.r.t. $x$,

$$\mathcal{W}_x(x,y) = \frac{\partial \mathcal{W}}{\partial x}.$$ (20)

$\mathcal{W}_x(x,y)$ gives the signed distance in pixels along the $x$-axis to the closest pixel in $\mathcal{B}(x,y) = 1$. This is a desirable property since it guides the contours towards a coarse segmentation of the tree stem given by the disparity map making the evolution of the contours robust to local minima in the image.

We now have the necessary components to define the external energy functionals. For the left image we combine the negative of the edge map, $\mathcal{I}_x(x,y)$, so that strong edges correspond to minima, and the negative of the contraction energy, $\mathcal{W}(x,y)$, so that the pixels where $\mathcal{B}(x,y) = 1$ are minimum. We also scale the contraction energy by the scalar $\kappa$ to balance it with the edge map so that the contraction energy becomes small as the contours approach the boundaries of $\mathcal{B}(x,y)$. For the right image, we simply use the negative of the edge map since the contraction force is derived from the disparity map that corresponds to the left image frame. The stem model corresponding to the right image will be influenced by the contraction energy through the stereo constraints introduced in Section 2.4. Formally, the external energy for the left and right images are defined as

$$\mathcal{E}_{\text{ext}}^L(x,y) = -\mathcal{I}_x^L(x,y) - \kappa \mathcal{W}(x,y),$$ (21a)
$$\mathcal{E}_{\text{ext}}^R(x,y) = -\mathcal{I}_x^R(x,y).$$ (21b)

The energy maps can be converted to horizontal force maps by taking their derivatives w.r.t. $x$. The gradient flow of the edge map, $\mathcal{G}(x,y)$, defined in the previous section is used as the force map for edges while $\mathcal{W}_x(x,y)$ is used for the contraction force,

$$\mathcal{F}^L(x,y) = \mathcal{G}^L(x,y) + \kappa \mathcal{W}_x(x,y),$$ (22a)
$$\mathcal{F}^R(x,y) = \mathcal{G}^R(x,y).$$ (22b)

We found $\kappa \in [0.05, 0.1]$ to be good values for scaling the contraction energy.

Figure 3 illustrates the progression from an input image to the force maps. Note that the capture range in $\mathcal{G}(x,y)$, Figure 3d, is far greater than the capture range in the force map $\mathcal{I}_{xx}(x,y)$, Figure 3c.

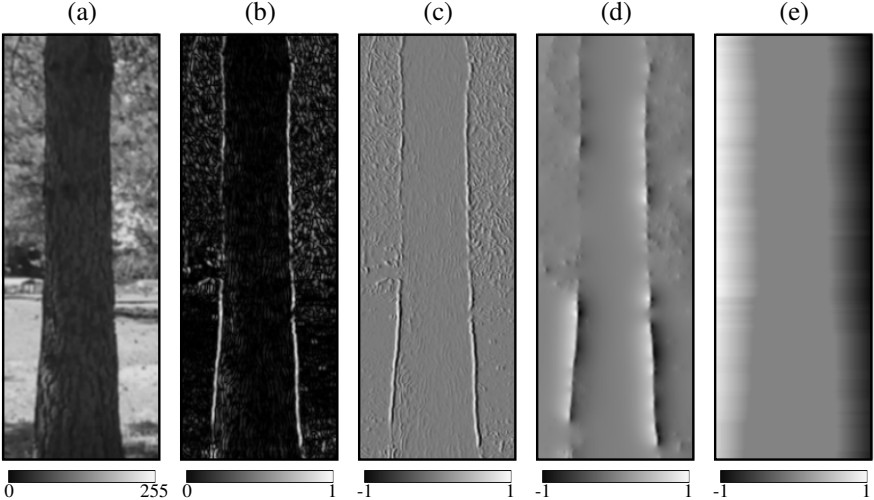

**Figure 3.** External energy (images are rescaled for visualization purposes): (**a**) Gray-scale image of a tree stem, $\mathcal{I}(x,y)$. (**b**) Edge map, $\mathcal{I}_x(x,y)$. (**c**) Force map $\mathcal{I}_{xx}(x,y)$. (**d**) Gradient flow force map, $\mathcal{G}(x,y)$. (**e**) Contraction force map, $\mathcal{W}_x(x,y)$.

### 2.2.3. External Forces on the Deformable Stem Model

In this section we show how the external force map is used to calculate the external forces acting on the Deformable Stem Model. Recall the parameters $\mathbf{x}$ and $\mathbf{r}$ used to specify an instance of the model. These parameters are used to construct the left and right contours of a tree stem, $\boldsymbol{\lambda}(s)$ and $\boldsymbol{\rho}(s)$, respectively. Since our external energy function, $\mathcal{E}_{\text{ext}}(x, y)$, is intended to be minimal at the boundary of the tree stem, our objective is to evolve the contours in such a way that each subsequent state results in a lower energy when integrated along the length of the contours. We can achieve this by subdividing the contours and integrating over the subintervals such that each subinterval corresponds to the control points in $\mathbf{x}$ and $\mathbf{r}$. The integrals along the subintervals are averaged and placed in the force vectors $\mathbf{f}_{\lambda} = (f_{\lambda,1}, f_{\lambda,2}, \dots, f_{\lambda,n})^{\mathsf{T}}$ and $\mathbf{f}_{\rho} = (f_{\rho,1}, f_{\rho,2}, \dots, f_{\rho,n})^{\mathsf{T}}$, then used to update $\mathbf{x}$ and $\mathbf{r}$. This is similar to the approach taken by Wang and Boyer [31] in the Active Geometric Shape Model.

We use $\mathbf{a} = (a_1, a_2, \dots, a_{n+1})^{\mathsf{T}}$ to represent the subintervals where each element is given by

$$a_1 = 0,  \tag{23a}$$

$$a_i = \frac{i}{n-1} - \frac{1}{2(n-1)}, \quad \forall \{i\}_2^n, \tag{23b}$$

$$a_{n+1} = 1. \tag{23c}$$

The individual force components can be computed by integrating over the gradient flow force function along the sub-intervals for the left and right contours:

$$f_{\lambda,i}^{(j)} = \frac{1}{a_{i+1} - a_i} \int_{a_i}^{a_{i+1}} \mathcal{F}^j\left(\boldsymbol{\lambda}^j(s)\right) \mathrm{d}s, \quad \forall \{i\}_1^n, j = \{L, R\}, \tag{24a}$$

$$f_{\rho,i}^{(j)} = \frac{1}{a_{i+1} - a_i} \int_{a_i}^{a_{i+1}} \mathcal{F}^j\left(\boldsymbol{\rho}^j(s)\right) \mathrm{d}s, \quad \forall \{i\}_1^n, j = \{L, R\}, \tag{24b}$$

where, again, $s \in [0, 1]$ is the parameterization. The components of the force vectors can be interpreted as the average force acting on the subinterval of the curve. So, if the part of the curve bounded by the subinterval is positioned to the left of a boundary in the energy function, then the force will be greater than 0; a force less than 0 indicates that the part of the curve is positioned to the right of the boundary. A force of zero means that the curve segment is balanced on the boundary.

The parameter vector $\mathbf{x}$, representing the control points for the curve of symmetry, can be updated to a state of lower energy by adding $\mathbf{x}$ and the vector sum of the forces acting on the left and right curves,

$$\mathbf{x}_{t+1}^{(j)} = \mathbf{x}_t^{(j)} + \mathbf{f}_{\lambda}^{(j)} + \mathbf{f}_{\rho}^{(j)}, \quad \forall j = \{L, R\}. \tag{25}$$

The parameter vector $\mathbf{r}$, representing the control points for the taper curve, can be updated by adding the vector with the forces acting on the right contour less the forces acting on the left,

$$\mathbf{r}_{t+1}^{(j)} = \mathbf{r}_t^{(j)} + \mathbf{f}_{\rho}^{(j)} - \mathbf{f}_{\lambda}^{(j)}, \quad \forall j = \{L, R\}. \tag{26}$$

To better understand these update equations, it is recommended that the reader uses Figure 3d,e to visualize different configurations of the left and right curves and how the update equations apply. Note that darker colors in the figures correspond to negative values while lighter colors correspond to positive values.

### 2.3. Internal Energy

We have yet to constrain the parameter set used to construct an instance of the Deformable Stem Model; the parameters **x** and **r** are controlled entirely by the external forces, some of which might not correctly represent the boundary of the tree stem. In this section, we introduce two biomechanically-inspired internal energy formulations that are simple analogies to how trees respond to physical forces in their environment.

2.3.1. Straightness Force

Trees have a natural tendency to grow straight. This is influenced by an internal control process called gravitropism, first recorded by Charles Darwin in 1880 [32]. Trees that grow against the gravity vector have the advantage that their weight is centered over the stem, thus the downward forces acting on the stem can be distributed. Trees also grow toward light sources through a weaker process, called phototropism, which can cause trees to lean or sweep toward openings in the canopy. Our objective here is to formulate an internal force for the stem model to encourage it to be straight while allowing for slight deviations to account for irregular stem forms.

We derive straightness energy using the normal equations for linear least squares regression. The energy of straightness corresponds to the sum of squared residuals from the best-fit line relating the parameter vector **x** and the arithmetic sequence **y** defined by Equation (3). Since the linear least squares model is undefined for a vertical line, we exchange the axes and define the design matrix on **y** by

$$\mathbf{Y} = \begin{pmatrix} \mathbf{1} & \mathbf{y} \end{pmatrix}, \tag{27}$$

and take **x** as the vector of dependent variables. The notation **1** denotes a vector of ones with the same dimension as **y**. Rearranging the normal equations for linear regression, we can get the predicted values for the best-fit line by

$$\hat{\mathbf{x}} = \Xi \mathbf{x}, \tag{28}$$

where $\Xi$ is the Hat matrix defined as

$$\Xi = \mathbf{Y}(\mathbf{Y}^\mathsf{T}\mathbf{Y})^{-1}\mathbf{Y}^\mathsf{T}. \tag{29}$$

The Hat matrix can also be used to compute the residuals, $\boldsymbol{\epsilon}$, by subtracting it from the identity matrix and right multiplying with the dependent vector,

$$\boldsymbol{\epsilon} = \underbrace{(\mathbf{I} - \Xi)}_{\mathbf{S}} \mathbf{x}. \tag{30}$$

We take the Euclidean norm of the residual vector as the straightness energy,

$$E_{\text{straightness}} = \left\| \mathbf{S}\mathbf{x} \right\|. \tag{31}$$

If the straightness energy is zero, then it is implied that the coordinates $\mathcal{X}$, defined in Section 2.1, are co-linear.

In order for the straightness energy to interact with other forces in the model, it is necessary to consider the force components that make up the scalar quantity describing the energy. This can be accomplished by discretization of the individual forces in the time domain and taking small steps in the direction of the forces. We can subtract $\alpha \mathbf{S}\mathbf{x}$ from **x**, where $\alpha \in [0, 1]$, to transition **x** to a state with lower energy. For example, if $\alpha = 0$, then $\mathbf{x} - \alpha \mathbf{S}\mathbf{x} = \mathbf{x}$, and if $\alpha = 1$, then $\mathbf{x} - \alpha \mathbf{S}\mathbf{x} = \Xi \mathbf{x} = \hat{\mathbf{x}}$. Using $\mathbf{x}_t$ to index **x** at time $t$, we have the following update rule for minimizing the straightness energy in the time domain,

$$\mathbf{x}_{t+1} = \mathbf{x}_t - \alpha \mathbf{S}\mathbf{x}_t. \tag{32}$$

Figure 4 illustrates an example of the internal straightness force acting on an instance of the Deformable Stem Model. The instance (a) has a straightness energy of 8.82. After 10 iterations with $\alpha = 0.1$ the energy of the stem model shown in (c) is decreased to 3.07.

(a)        (b)        (c)

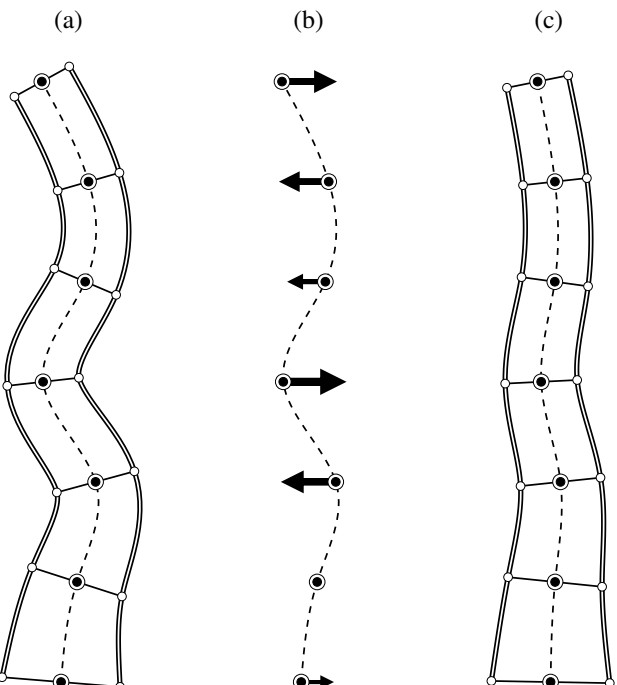

**Figure 4.** Internal straightness forces: (**a**) An instance of the Deformable Stem Model exhibiting an unlikely form. (**b**) Vectors showing the direction and magnitude of the internal straightness forces, **Sx**, scaled up here for visualization purposes. (**c**) The result of instance (**a**) after 10 iterations at $\alpha = 0.1$ with only the internal straightness forces acting on the stem.

### 2.3.2. Monotonic Radii Force

In 1893, Metzger [33] provided an biomechanical explanation for the taper of tree stems. Metzger hypothesized that trees distribute radial growth along their stem to uniformly counterbalance the mechanical stress acting on the tree caused by lateral forces, e.g., wind. A beam of uniform resistance to these forces is a cubic paraboloid, a geometric shape that closely resembles that of a tree stem. In reality, trees have more complicated taper functions, as indicated in numerous studies [34–36]. Here, we enforce a simple constraint on the stem model to encourage the radius parameters to be monotonically decreasing with height to allow for variations among species and individuals.

Let **A** be a backwards difference matrix. We can construct this matrix by subtracting an $n \times n$ identity matrix, $\mathbf{I}_n$, by a super-diagonal shift matrix, $\mathbf{U}_n$, of the same shape,

$$\mathbf{A} = \mathbf{U}_n - \mathbf{I}_n \,. \tag{33}$$

If we take the matrix-vector product **Ar**, then the positive values in the resulting vector indicate that the corresponding element in the radius vector, **r**, has a larger subsequent element, i.e., the projected width of the stem is increasing with height, which is generally not how trees taper. Negative values indicate that the width is decreasing, while zero indicates a constant taper. Our objective here is to formulate an energy that is zero when the radius vector of the stem is monotonically decreasing with height, and positive otherwise. We can achieve this by first passing the matrix-vector product **Ar** through a linear rectifier to truncate all negative values to zero. The rectifier is a vector-valued function, $\Psi : \mathbb{R}^n \to \mathbb{R}^n$, defined as

$$\Psi(\mathbf{p}) = \begin{pmatrix} \max(0, p_1) & \max(0, p_2) & \dots & \max(0, p_n) \end{pmatrix}^\mathsf{T}, \tag{34}$$

where **p** is a placeholder for the product of **Ar**. Next, we average bidirectional differences of increasing values with the operation $-\frac{1}{2}\mathbf{A}^\mathsf{T}\Psi(\mathbf{Ar})$. The sign and magnitude of the elements in the resulting vector guide the movement of the control points for the taper curve to a state of lower energy, where the energy, $E_{\text{taper}}$, is given by

$$E_{\text{taper}} = \left\| -\frac{1}{2}\mathbf{A}^\mathsf{T}\Psi(\mathbf{Ar}) \right\|, \tag{35}$$

We can discretize the movement of **r** in the time domain by introducing the step size parameter, $\beta$, leading to the following update rule for minimizing taper energy:

$$\mathbf{r}_{t+1} = \mathbf{r}_t - \beta\left(\frac{1}{2}\mathbf{A}^\mathsf{T}\,\Psi(\mathbf{Ar}_t)\right). \tag{36}$$

Figure 5 shows the internal taper force acting on an instance of the Deformable Stem Model. The instance shown in (a) has an energy of 2.34. The resulting model following 10 iterations with $\beta = 0.1$, instance (c) in the figure, has an energy of 0.72.

(a)                        (b)                       (c)

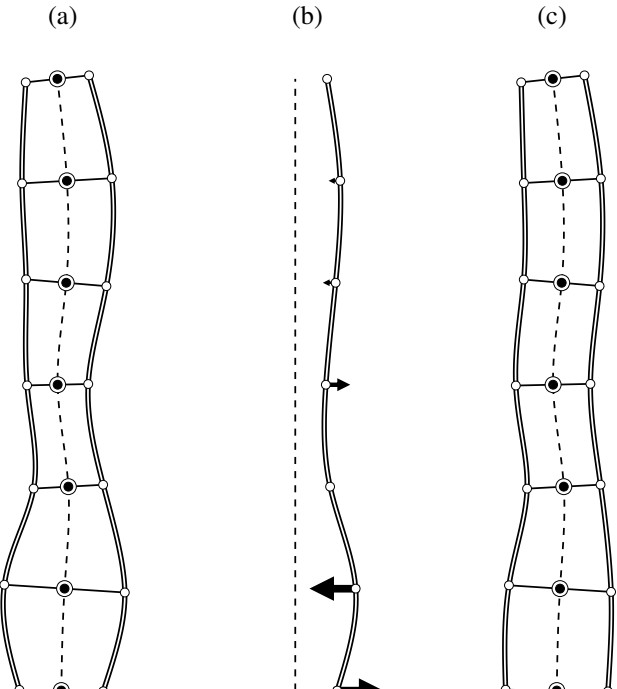

**Figure 5.** Internal taper forces: (**a**) An instance of the Deformable Stem Model exhibiting an unlikely taper function. (**b**) Vectors showing the direction and magnitude of the internal taper forces, scaled up here for visualization purposes. (**c**) The result of instance (**a**) after 10 iterations at $\beta = 0.1$ with only the internal taper forces acting on the stem.

*2.4. Stereo Energy*

In this section we introduce the two final energy formulations. The energy terms presented here create a connection between the models in the left and right images. Specifically, information regarding the contours in one image is used to localize the contours in the other. This is especially useful when there are desirable features missing in one image that are present in the other.

We define a stereo smoothness constraint on the curves of symmetry in the left and right image as

$$E_{\text{stereo-sym}} = \frac{1}{2}\left(\frac{\mathrm{d}\chi^L}{\mathrm{d}x} - \frac{\mathrm{d}\chi^R}{\mathrm{d}x}\right)^2. \tag{37}$$

This ensures that the disparities between the curves vary smoothly. $E_{\text{stereo-sym}} = 0$ means that the derivatives of the curves w.r.t. $x$ are exactly the same, while $E_{\text{stereo-sym}} > 0$ means the disparities diverge somewhere along the length of the curves. Since $\chi(s)$ is a spline interpolation of the parameter vector **x**, we can define force update equations for the left and right model parameters as

$$\mathbf{x}_{t+1}^{(L)} = \mathbf{x}_t^{(L)} + \mu\left(\mathbf{Dx}_t^{(R)} - \mathbf{Dx}_t^{(L)}\right),\tag{38a}$$

$$\mathbf{x}_{t+1}^{(R)} = \mathbf{x}_t^{(R)} + \mu\left(\mathbf{Dx}_t^{(L)} - \mathbf{Dx}_t^{(R)}\right),\tag{38b}$$

where $\mu$ is the step size parameter and **D** is a finite difference matrix. We use the step size parameter to make a soft constraint on the differences. This allows the disparity between the curves to vary slightly when the tree stem is not exactly vertically aligned with the image plane.

The second stereo constraint forces the taper curve of the left and right model to be proportional,

$$E_{\text{stereo-tap}} = \tau^L(s) - \gamma\tau^R(s)\tag{39}$$

where $\gamma$ is some scalar that, when multiplied by one of the taper curves, minimizes their differences. Since the two taper profiles are projections of essentially the same face of the stem offset by a small baseline, we expect their taper functions to be identical. However, depending on the angle at which the stem is positioned w.r.t. the optical axes of the camera, the distance from the stem to the left camera can be different than the distance from the stem to the right camera. Thus, we allow the scale of the taper function to differ. This constraint can be converted to force update equations and applied to the left and right radius parameter vectors by first taking the average of the two vectors and then adding the sum of their difference divided by $2n$,

$$\mathbf{r}_{t+1}^{(L)} = \frac{1}{2}\left(\mathbf{r}_t^{(L)} + \mathbf{r}_t^{(R)}\right) + \frac{1}{2n}\sum_i^n\left(r_{t,i}^{(L)} - r_{t,i}^{(R)}\right),\tag{40a}$$

$$\mathbf{r}_{t+1}^{(R)} = \frac{1}{2}\left(\mathbf{r}_t^{(L)} + \mathbf{r}_t^{(R)}\right) + \frac{1}{2n}\sum_i^n\left(r_{t,i}^{(R)} - r_{t,i}^{(L)}\right).\tag{40b}$$

*2.5. Optimization*

In this section we show how we minimize the external, internal and stereo energy of the Deformable Stem Model via gradient descent. First, the parameters for the model, $\{\mathbf{x}, \mathbf{r}\}$, are initialized according to the width of the image domain $\Phi_i$. We set all values of **x** and **r**, for both the left and right models, equal to $\frac{w}{2}$, where $w$ is the number of pixel columns in $\Phi_i$. Thus, the left and right contours, when computed using Equation (1), are initialized at the left and right edges of the bounding box. We also initialize **y** with Equation (3) where $h$ in the equation is set to the number of pixel rows in $\Phi_i$. The matrices **S**, **D**, and **A** are constants, so they are initialized and stored. The external force maps for the left and right images are computed using Equation (22).

Next, we concatenate all the force update equations for the left and right parameter vectors $\mathbf{x}^{(L)}$ and $\mathbf{x}^{(R)}$,

$$\mathbf{x}_{t+1}^{(L)} = \mathbf{x}_t^{(L)} - \underbrace{\alpha\mathbf{Sx}_t^{(L)}} + \underbrace{\mu\left(\mathbf{Dx}_t^{(R)} - \mathbf{Dx}_t^{(L)}\right)} + \underbrace{\mathbf{f}_\gamma^{(L)} + \mathbf{f}_\rho^{(L)}},\tag{41a}$$

$$\mathbf{x}_{t+1}^{(R)} = \mathbf{x}_t^{(R)} - \underbrace{\alpha\mathbf{Sx}_t^{(R)}}_{\text{internal}} + \underbrace{\mu\left(\mathbf{Dx}_t^{(L)} - \mathbf{Dx}_t^{(R)}\right)}_{\text{stereo}} + \underbrace{\mathbf{f}_\gamma^{(R)} + \mathbf{f}_\rho^{(R)}}_{\text{external}},\tag{41b}$$

and for the parameter vectors $\mathbf{r}^{(L)}$ and $\mathbf{r}^{(R)}$,

$$
\mathbf{r}_{t+1}^{(L)} = \mathbf{r}_t^{(L)} - \underbrace{\beta\left(\frac{1}{2}\mathbf{A}^\mathsf{T}\Psi\left(\mathbf{A}\mathbf{r}_t^{(L)}\right)\right)}_{\text{internal}}
$$

$$
+ \underbrace{\frac{1}{2}\left(\mathbf{r}_t^{(L)} + \mathbf{r}_t^{(R)}\right) + \frac{1}{2n}\sum_i^n\left(r_{t,i}^{(L)} - r_{t,i}^{(R)}\right)}_{\text{stereo}} + \underbrace{\mathbf{f}_\rho^{(L)} - \mathbf{f}_\gamma^{(L)}}_{\text{external}}, \tag{42a}
$$

$$
\mathbf{r}_{t+1}^{(R)} = \mathbf{r}_t^{(R)} - \underbrace{\beta\left(\frac{1}{2}\mathbf{A}^\mathsf{T}\Psi\left(\mathbf{A}\mathbf{r}_t^{(R)}\right)\right)}_{\text{internal}}
$$

$$
+ \underbrace{\frac{1}{2}\left(\mathbf{r}_t^{(L)} + \mathbf{r}_t^{(R)}\right) + \frac{1}{2n}\sum_i^n\left(r_{t,i}^{(R)} - r_{t,i}^{(L)}\right)}_{\text{stereo}} + \underbrace{\mathbf{f}_\rho^{(R)} - \mathbf{f}_\gamma^{(R)}}_{\text{external}}. \tag{42b}
$$

We iterate these equations and during each iteration we construct a cubic spline interpolant through $\{\mathbf{x}^{(L)}, \mathbf{r}^{(L)}\}$ and $\{\mathbf{x}^{(R)}, \mathbf{r}^{(R)}\}$ in order to compute the edge contours $\{\boldsymbol{\lambda}^L(s), \boldsymbol{\rho}^L(s)\}$, and $\{\boldsymbol{\lambda}^R(s), \boldsymbol{\rho}^R(s)\}$. This is necessary for calculating the external force vectors—the last terms in Equations (41) and (42). After each iteration, we check for convergence, where the convergence value is given by

$$
\epsilon = \left\|\mathbf{x}_{t+1}^{(L)} - \mathbf{x}_t^{(L)}\right\| + \left\|\mathbf{r}_{t+1}^{(L)} - \mathbf{r}_t^{(L)}\right\| + \left\|\mathbf{x}_{t+1}^{(R)} - \mathbf{x}_t^{(R)}\right\| + \left\|\mathbf{r}_{t+1}^{(R)} - \mathbf{r}_t^{(R)}\right\|. \tag{43}
$$

The iteration is terminated when $\epsilon$ is less than some threshold, e.g., 0.1. We do this to avoid issues with numerical instability when the parameters oscillate around a minimum. Once the model parameters have converged, we use the stereo calibration parameters to reconstruct the model using Equations (7)–(10).

Refer to Figure 6 for a diagram showing the flow of operations in the algorithm. We have referenced the core equations used in the text boxes between the visual representations. Figure 7 illustrates curve evolution during optimization of the Deformable Stem Model on six example trees. For each example, we show 5 snap-shots of the contours corresponding to 1, 5, 15, 30, and 50 iterations, respectively. The left-most image in each subfigure shows the position of the left and right edge contour following the first update after initialization based on the given bounding box. The right-most image shows the edge contours after convergence. The images in between are intermediate time steps. We also alternate left and right stereo images, so images 1, 3, 5 are from the left camera and 2 and 4 are from the right camera.

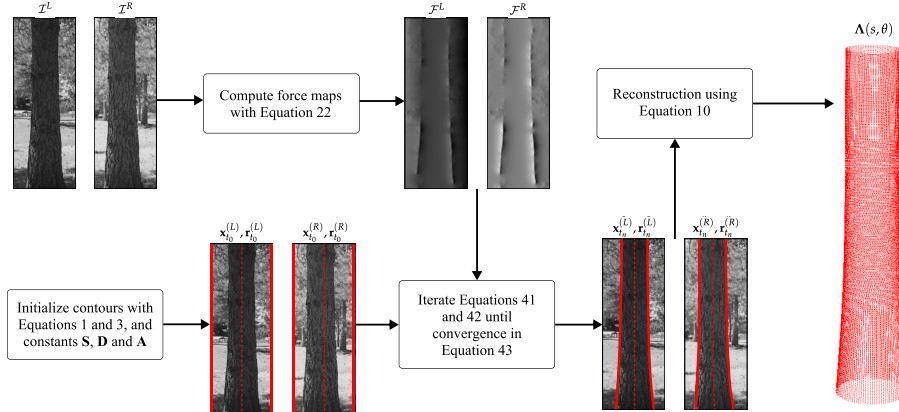

**Figure 6.** Diagram showing the flow of operations in the Deformable Stem Model algorithm.

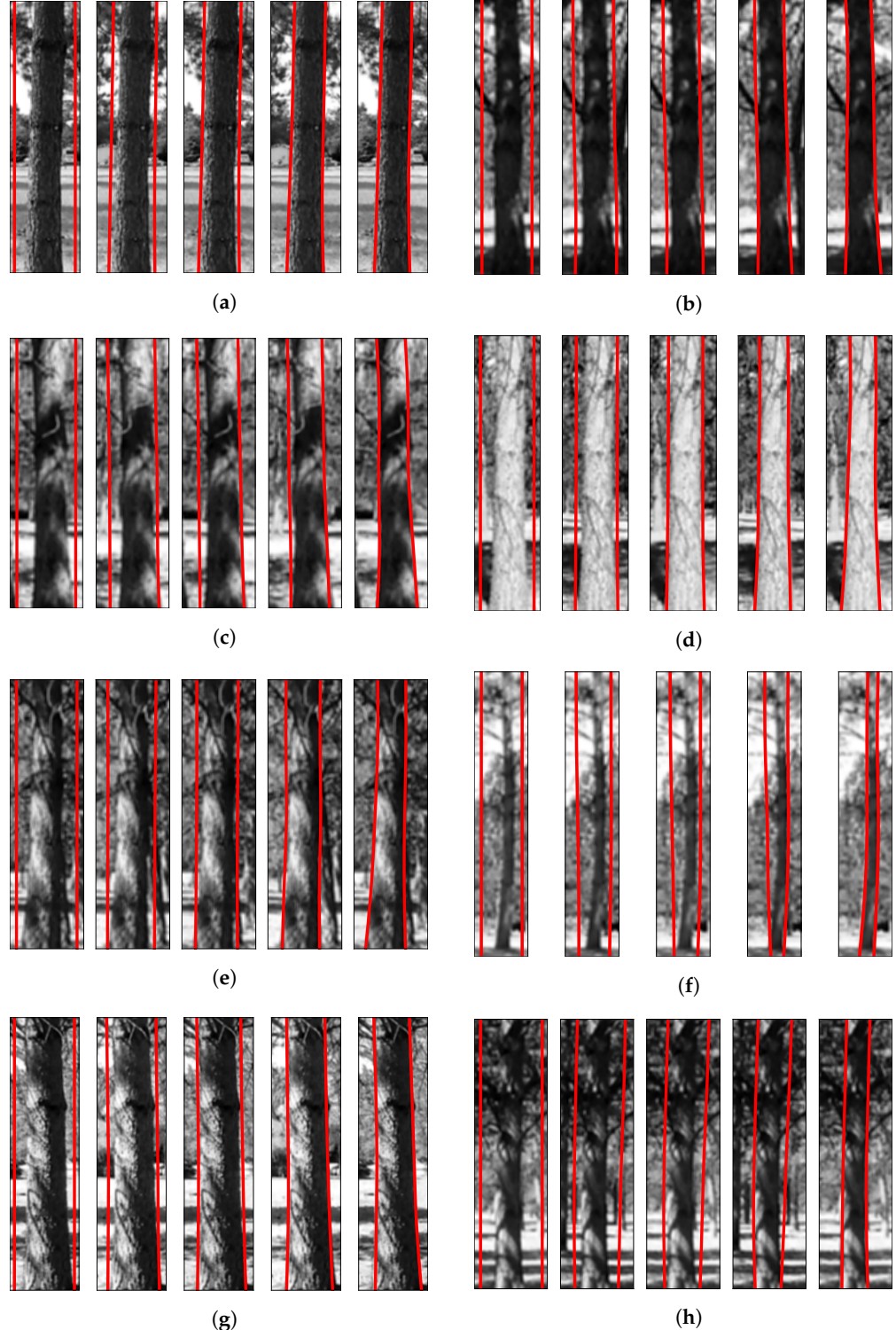

**Figure 7.** Six examples of optimization of the Deformable Stem Model on real images. (**a**) Homogeneously illuminated stem with clear edges. (**b**) Discontinuous edges due to branching. (**c**) Harsh shadows crossing the stem. (**d**) Fully illuminated stem with low contrast to background. (**e**) Partially occluded stem in the background. (**f**) Curved stem at maximum working distance of camera. (**g**) Partially illuminated stem. (**h**) Low contrast stem with branching.

## 3. Application to Diameter Measurement

### 3.1. Materials

We tested the Deformable Stem Model using a custom-built stereo camera with a 32 cm baseline. The cameras were operated at VGA resolution (640 × 480) with a focal

length of 3.0 mm. The cameras were calibrated for intrinsic and extrinsic parameters following methods presented in [37], and the images were stereo-rectified and row-aligned. We captured 7 stereo pairs within a sparse ponderosa pine (*Pinus ponderosa* Douglas ex Lawson) forest in western Montana, each photograph containing 2 to 4 trees within 20 m of the camera, for a total of 21 trees. We recorded a caliper measurement for each tree across the face of the stem pointing toward the camera at breast height (1.37 m) and the distance from the plane of the left camera imaging sensor to the center of each tree at breast height. Table 1 shows the mean, standard deviation, minimum, and maximum values for the collection data.

**Table 1.** Univariate statistics of diameter observations. Units are in centimeters.

| Observation | Mean | STD | Min | Max |
|---|---|---|---|---|
| Distance ($z$) | 1017.45 | 458.85 | 350.52 | 1813.56 |
| Breast ht. ($y$) | 137.16 | 0.00 | 137.16 | 137.16 |
| Diameter ($d$) | 35.09 | 4.88 | 26.00 | 42.30 |

We also captured a second stereo pair at each plot after wrapping each tree with orange flagging at the height of measurement so that the exact point of measurement along the stem could be converted to a pixel location for validation purposes. We minimized the energy of the Deformable Stem Model on each tree using the parameter configuration shown in Table 2.

**Table 2.** Deformable Stem Model hyperparameters used in validation experiment.

| Hyper-Parameter | Description | Value |
|---|---|---|
| $n$ | number of control points | 5 |
| $\alpha$ | internal straightness force step size | 0.1 |
| $\beta$ | internal taper force step size | 0.1 |
| $\kappa$ | contraction force scaling parameter | 0.1 |
| $\mu$ | stereo smoothness constraint parameter | 0.3 |

*3.2. Analysis*

First, we tested the accuracy of diameter measurements without using the camera calibration parameters to reconstruct the stem model. We do this to assess how potential errors in camera calibration contribute to the measurement accuracy. We use the notation $\tau(y)$ to denote the taper function at the observed breast height measurement position $y$. The taper function returns the number of pixels from the center of the tree to the edge, so scaling by 2 gives the number of pixels across the entire stem. The estimated diameter at breast height is given by

$$\hat{d} = \frac{2\tau(y)z}{f}, \tag{44}$$

where $z$ is the observed distance from the camera to the tree and $f$ is the focal length. This implies that

$$\hat{d} \propto 2\tau(y)z. \tag{45}$$

Thus, we can relate the observed diameter measurements, $d$, to the proportional estimates, $2\tau(y)z$, without using the calibration parameter $f$. We use the linear regression parameters, $\beta_0$ and $\beta_1$ to compute the estimated diameter,

$$\hat{d}(z, y) = \frac{2\tau(y)z - \beta_0}{\beta_1}. \tag{46}$$

Next, we estimate the diameter using the stereo disparity from the left and right stem models and the calibration parameters to estimate the distance to the tree stem. This estimate is not considered completely automatic since we use the observed height of measurement position $y$. The estimator for the distance is given by,

$$\hat{z}(y) = f b \delta(y), \tag{47}$$

where $b$ is the baseline distance between the cameras in meters and $\delta(s)$ is the inverse disparity function defined in Equation (7). Plugging $\hat{z}(y)$ into Equation (44) yields

$$\hat{d}(\hat{z}, y) = \frac{2\tau(y)\hat{z}(y)}{f}. \tag{48}$$

Note that this equation can be simplified so that the variable $f$ appears in the numerator and denominator. However, we keep the equation as is to indicate that $z$ is a function of a specific height argument.

For the final diameter estimator, we replace the known height position variable, $y$, with an estimated measurement height. We used RANSAC [38] to extract the ground plane from the point cloud generated with the disparity map and used the best-fit plane as a reference to calculate breast height [39]. The estimator for the measurement height, $\hat{y}$, corresponding to the best-fit plane describing the ground, is defined as

$$\hat{y} = \underset{s \in [0,1]}{\text{argmin}} \left| \hat{\mathbf{n}}(\boldsymbol{X}(s) - \boldsymbol{\mu}) - 1.37 \right|, \tag{49}$$

where $\boldsymbol{X}(s)$ is the back-projected curve of symmetry defined in Equation (8), $\hat{\mathbf{n}} \in \mathbb{R}^3$ is the unit normal vector of the plane, $\boldsymbol{\mu} \in \mathbb{R}^3$ is an arbitrary point on the plane and 1.37 is the distance above the ground to breast height in meters. The final diameter estimator is given by

$$\hat{d}(\hat{z}, \hat{y}) = \frac{2\tau(\hat{y})\hat{z}(\hat{y})}{f}. \tag{50}$$

For each estimator we calculated the root mean squared error (RMSE), defined as

$$\left[ \frac{1}{N} \sum_{i}^{N} \left( \hat{\theta} - \theta \right)^2 \right]^{\frac{1}{2}}, \tag{51}$$

the bias on the estimate, defined as

$$\frac{1}{N} \sum_{i}^{N} \left( \hat{\theta} - \theta \right). \tag{52}$$

and the mean absolute error (MAE), given by

$$\frac{1}{N} \sum_{i}^{N} \left| \hat{\theta} - \theta \right|. \tag{53}$$

## 4. Results and Discussion

### 4.1. Diameter Measurement

Figure 8a illustrates the relationship between the ground truth diameter, $d$, and the projected width in pixels, $2\tau(y)$, scaled by the ground truth distance, $z$. Figure 8b shows the relationship between $z$ and $\hat{z}(\cdot)$, where the open circles in the figure correspond to $\hat{z}(y)$. As the figure suggests, any existing calibration errors are minuscule. The estimated diameters according to Equation (50) are plotted against the observed diameters in Figure 8c.

The residuals of the estimates are plotted against the observed distance in Figure 8d to show that the errors do not appear to be correlated with distance.

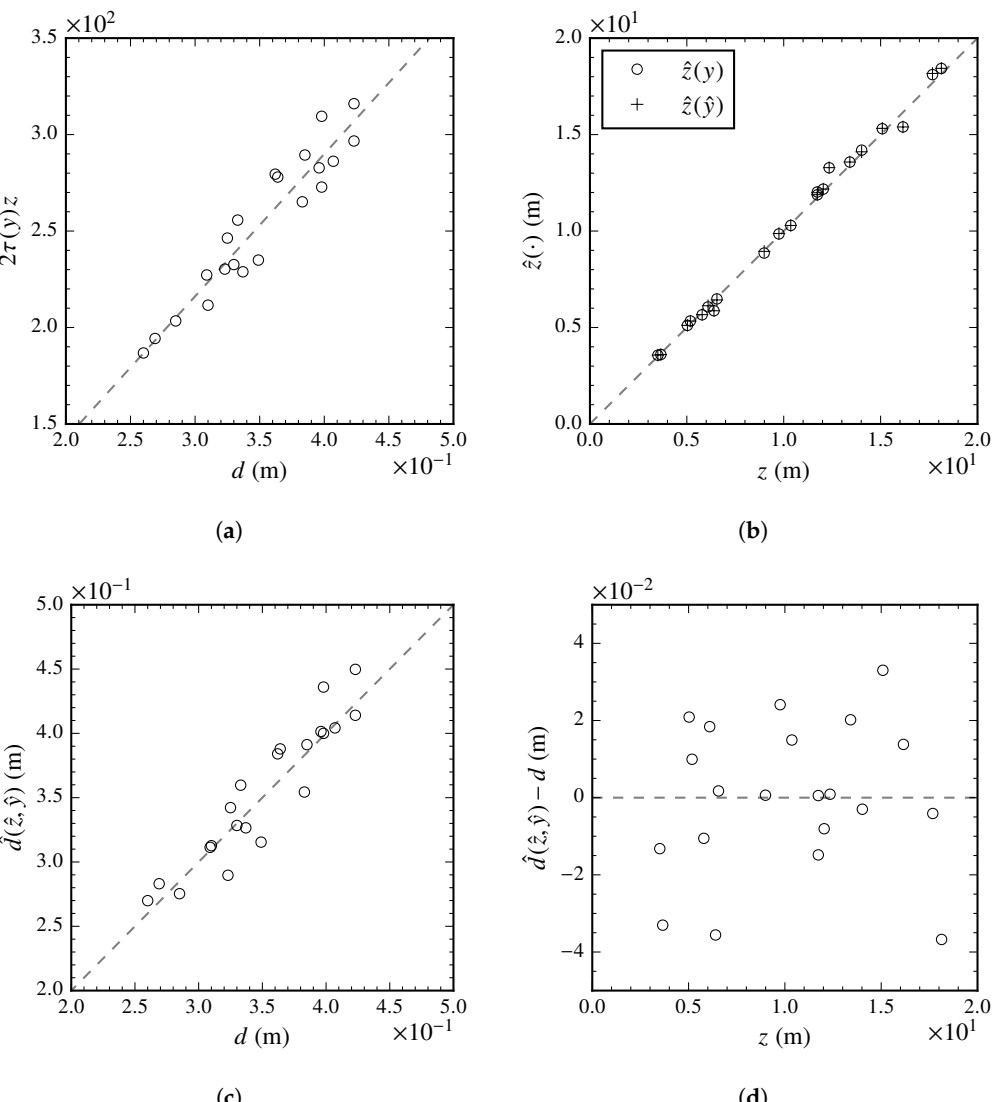

**Figure 8.** Diameter estimation results. (**a**): Diameter vs. projected width, $2\tau(y)$, at observed breast height, $y$, scaled by observed distance, $z$. The gray dashed line is the best-fit linear model. (**b**): Observed distance, $z$, vs. predicted distance at observed breast height, $\hat{z}(y)$, and predicted distance at predicted breast height, $\hat{z}(\hat{y})$. (**c**): Observed diameter, $d$, vs. predicted diameter at predicted distance and predicted breast height, $\hat{d}(\hat{z}, \hat{y})$. (**d**): Observed distance, $z$ vs. diameter residuals, $\hat{d}(\hat{z}, \hat{y}) - d$.

The RMSE, bias, and MAE of the estimates are presented in Table 3. We focus our discussion on RMSE as it penalizes large errors more than MAE. The RMSE for the calibration-free diameter estimate, $\hat{d}(z, y)$, was 1.61 cm with a small bias of 0.02 cm. We consider this estimate to have the highest potential accuracy among all other estimates assuming error-free distance observations. Using the camera calibration parameters to estimate the distance increased the RMSE of the diameter estimate by 0.15 cm and the bias by 0.11 cm. This marginal increase in error suggests that our camera calibration parameters are sufficiently accurate. The final diameter estimate, using both the estimated distance and measurement height, had an RMSE of 1.94 cm, 0.33 cm greater than the calibration-free estimator, and slightly overestimated the ground truth mean by 0.3 cm. In comparison to other recent studies that focus only on non-contact diameter measurement at breast height, our RMSE is within the range of their results: 7.9 cm [40], 2.71 cm [41], 1.28–2.57 cm [42], and 1.55 cm [43].

**Table 3.** RMSE, bias, and MAE for the distance, breast height, and diameter estimators. The bold values show the results for the fully automatic diameter estimator.

| Estimator | | Equation | Mean (cm) | STD (cm) | RMSE (cm) | Bias (cm) | MAE (cm) |
|---|---|---|---|---|---|---|---|
| Distance | $\hat{z}(y)$ | Equation (47) | 1023.34 | 468.38 | 33.00 | 5.89 | 23.26 |
| | $\hat{z}(\hat{y})$ | Equation (47) | 1023.82 | 468.47 | 33.27 | 6.38 | 23.10 |
| Breast ht. | $\hat{y}$ | Equation (49) | 135.03 | 5.85 | 6.10 | −2.13 | 4.87 |
| Diameter | $\hat{d}(z,y)$ | Equation (46) | 35.11 | 5.23 | 1.61 | 0.02 | 1.39 |
| | $\hat{d}(\hat{z},y)$ | Equation (48) | 35.22 | 5.51 | 1.76 | 0.13 | 1.49 |
| | $\hat{d}(\hat{z},\hat{y})$ | Equation (50) | **35.39** | **5.42** | **1.94** | **0.30** | **1.55** |

*4.2. Run Time Analysis*

The algorithm was implemented in the C++ programming language using the CUDA parallel computing platform for interfacing with parallelization elements of the GPU [44]. We performed a worst-case run time analysis on 25 images of trees ranging from 2–56 kilopixels (kp) in size. We considered this analysis a worst-case evaluation as we set the number of iterations for the optimization step to 100. This is approximately double the number of iterations required for convergence when we tested the algorithm on the 21 trees in the diameter validation experiment. In Table 4, we show the average time in milliseconds (ms) for the major components of the algorithm configured according to Table 2. The table also summarizes the time required for allocating GPU memory and uploading data to the GPU. These tests were performed using a NVIDIA GeForce GTX 780M GPU.

**Table 4.** Run time results for various components of the algorithm.

| Size (kp) | 0–10 | 10–20 | 20–30 | 30–40 | 40–50 |
|---|---|---|---|---|---|
| **Upload (ms)** | 0.04 | 0.04 | 0.07 | 0.08 | 0.10 |
| **Allocate (ms)** | 0.16 | 0.24 | 0.40 | 0.58 | 0.77 |
| **Gradient flow (ms)** | 0.88 | 1.12 | 1.91 | 2.70 | 3.57 |
| **Contraction force (ms)** | 0.24 | 0.32 | 0.44 | 0.58 | 0.70 |
| **Optimization-100 iterations (ms)** | 2.42 | 3.20 | 4.18 | 4.77 | 5.18 |
| **Total (ms)** | 3.74 | 4.93 | 6.99 | 8.71 | 10.30 |
| **Frames per second** | 267.34 | 202.85 | 142.99 | 114.80 | 97.05 |

The average run time across all test images was 5.57 ms (179.53 frames per second). In the table, we do not include the run time of the stereo-matching algorithm that is required for computing the contraction force since it depends heavily on the choice of stereo-matching algorithm and implementation. The implementation we used [30], can perform stereo matching for a VGA resolution image in 11 ms (90 fps) on the GTX 780M GPU. This increases the run time of our algorithm to 16.57 ms and decreases the frame rate to 60 fps. Furthermore, this analysis summarizes the run time for a single tree. If we expect to have on average 10 tree detections at any given time, then the run time, including the time required for stereo matching, will increase to approximately 66 ms (15 fps).

**5. Conclusions**

In this paper, we demonstrated how 2D projections of tree stems in stereo images can be automatically reconstructed into 3D models in real time. We show how dendrometric attributes can be calculated from the models and provide an accuracy assessment of diameter measurements for 21 tree stems. The algorithm presented is based on the active contour model, a well-developed computer vision framework for boundary localization. The algorithm runs in real-time on a modest GPU and is intended to be used during forest harvesting operations for diameter measurement and stem form analysis of standing trees.

There are many attributes of a tree stem that influence its value. Stem size and shape are among the most important variables and can also be used to make optimal decisions regarding bucking positions and milling routines [45,46]. The algorithm presented in this paper can provide 3D stem models for product value optimization. Although our approach only provides caliper measurements for the sub-canopy portion of the stem, the shape of the stem is calculated in 3D. Since these models can be generated in real-time, they can be used in harvesting operations to make optimal bucking decisions and to calculate the value of the stem.

Research has also been conducted to relate standing tree shape to the spatial distribution of internal wood characteristics. Constant et al. [47] presented an analysis of poplar trees that suggests a strong relationship between local tree slope and the position of the pith in relation to the center of the stem. They also found that the concentration of reaction wood is related to the local slope of the stem. Since reaction wood and the eccentricity of annual rings are directly related to the quality and price of the products derived from trees, 3D models like those provided by our algorithm can be useful in determining the value of logs prior to harvesting.

The data used in our diameter validation experiment were collected in a park-like forest where the trees were uniformly distributed and similar in age class. We have also tested our algorithm in a natural forest where trees are clustered and vary in age class. Although our algorithm performs well in natural forests, it is unclear what density of trees or level of understory vegetation will limit the performance of the algorithm.

A limitation of this work is that it is based on only two unique views of the stem as opposed to many in the case of MVS-SFM, essentially providing caliper measurements of the stem. We do not, however, consider this to be a major limitation as caliper and d-tape measurements have been shown to be statistically similar [48]. Another limitation is that our method does not provide tree height, which is an important dendrometric variable. We recommend that future research in this arena consider the use of fish-eye or hemispherical lenses to capture the upper portions of tree stems.

The analysis presented in this paper did not provide insight to the maximum distance a tree can be from the camera before errors render the estimates unreliable. We found that diameter errors do not increase with distance for trees up to 20 m from the camera, but further distances might be possible. Increasing the resolution of the camera will certainly increase the working distance and accuracy of contour localization; however, at the cost of increased processing time. An alternative is to increase the baseline between the cameras; however, this will increase the minimum working distance and trees close to the camera might not appear in both stereo images. Further research is needed to study the relationship between camera configuration, resolution, processing time and accuracy.

## 6. Patents

US Patent No. US011481972B2: Method of performing dendrometry and forest mapping.

**Author Contributions:** Conceptualization, L.A.W. and W.C.; methodology, L.A.W. and W.C.; software, L.A.W.; validation, L.A.W. and W.C.; formal analysis, L.A.W.; investigation, L.A.W. and W.C.; resources, L.A.W.; data curation, L.A.W.; writing—original draft preparation, L.A.W.; writing—review and editing, L.A.W. and W.C.; visualization, L.A.W.; supervision, W.C.; project administration, W.C.; funding acquisition, W.C. All authors have read and agreed to the published version of the manuscript.

**Funding:** This research was funded by the U.S. Forest Service National Technology and Development Program under contract number 16CS-1113-8100-017.

**Data Availability Statement:** Data are contained within the article.

**Conflicts of Interest:** The authors declare no conflict of interest.

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
