# Peer review of "A Deformable Shape Model for Automatic and Real-Time Dendrometry"

_forests, doi:10.3390/f14122299_

Round 1

Reviewer 1 Report

Comments and Suggestions for Authors

Dear Authors,

I want to thank the authors of this research first of all. Their case study details the steps involved in creating and getting ready a fairly sophisticated algorithm in accordance with forestry requirements. At this point, I'd like to make clear that while this article's explanation of a portion of a patent-registered piece of software is comparable to the authors' earlier works, it differs from them in terms of content. It is undeniable that an already-in-use algorithm will significantly advance the field of forestry research. Neither the article's language nor its content can be faulted. There is only one aspect of the study that can be considered as improvable. This is during the accuracy analysis step, 21 tree trunks were examined. Although the study may be improved in this area, the results are still deemed satisfactory because a remarkable amount of work was put into developing the algorithm. In addition, a few recommendations might be made from the perspective of the reader to make it simpler for those who are interested. A graphical representation of the algorithmic path and an indication of the work's mainstages will improve comprehension at this point. A graphical abstract would make the article appear more appealing at first sight.  

Best regards.

Reviewer 2 Report

Comments and Suggestions for Authors

In this study, an stereo image based algorithm is introduced for measuring tree trunk diameter; it is stated that the algorithm can automatically estimate the diameter of trees up to 20 meters away with a 5.52% margin of error

The methodology introduced in this study exhibits potential for selective implementation across diverse forestry operations. As delineated in the concluding remarks, the technique may not consistently manifest optimal outcomes in natural forest ecosystems characterized by stratified arboreal forest layers or understory vegetative density. An empirical validation of the algorithm under authentic forest conditions, beyond park-like settings, would enhance the robustness and broader applicability of our findings.

However, despite some limitations, this approach is valuable as a simple method that can be integrated into mobile/etc.. applications for forestry operations.

PS: double 'the' in line 541

Reviewer 3 Report

Comments and Suggestions for Authors

The idea is good. However, I are very suprised that the proposed method is not be compared in experiments. Then, how to present the advantages of this work.  There is no researcher on this area?
